# Unraveling the Impact of Environmental Factors and Evolutionary History on Species Richness Patterns of the Genus *Sorbus* at Global Level

**DOI:** 10.3390/plants14030338

**Published:** 2025-01-23

**Authors:** Yujia Pan, Chenlong Fu, Changfen Tian, Haoyue Zhang, Xianrong Wang, Meng Li

**Affiliations:** Co-Innovation Center for Sustainable Forestry in Southern China, College of Life Sciences, Nanjing Forestry University, Nanjing 210037, China; panyujia8@163.com (Y.P.); chenlongfu1219@njfu.edu.cn (C.F.); t17895012865@163.com (C.T.); mimanchi983@163.com (H.Z.)

**Keywords:** *Sorbus*, species richness patterns, environmental factors, diversification rate

## Abstract

Understanding the drivers of species richness patterns is a major goal of ecology and evolutionary biology, and the drivers vary across regions and taxa. Here, we assessed the influence of environmental factors and evolutionary history on the pattern of species richness in the genus *Sorbus* (110 species). We mapped the global species richness pattern of *Sorbus* at a spatial resolution of 200 × 200 km, using 10,652 specimen records. We used stepwise regression to assess the relationship between 23 environmental predictors and species richness and estimated the diversification rate of *Sorbus* based on chloroplast genome data. The effects of environmental factors were explained by adjusted R^2^, and evolutionary factors were inferred based on differences in diversification rates. We found that the species richness of *Sorbus* was highest in the Hengduan Mountains (HDM), which is probably the center of diversity. Among the selected environmental predictors, the integrated model including all environmental predictors had the largest explanatory power for species richness. The determinants of species richness show regional differences. On the global and continental scale, energy and water availability become the main driving factors. In contrast, climate seasonality is the primary factor in the HDM. The diversification rate results showed no significant differences between HDM and non-HDM, suggesting that evolutionary history may have limited impact on the pattern of *Sorbus* species richness. We conclude that environmental factors play an important role in shaping the global pattern of *Sorbus* species richness, while diversification rates have a lesser impact.

## 1. Introduction

Understanding the drivers of global biodiversity is crucial for developing effective conservation strategies [1]. The environment is generally considered the main driver of species richness patterns, influencing species distribution, phylogeny, and evolution [2]. Consequently, global species richness is unevenly distributed, with notable biodiversity hotspots [3,4,5]. Patterns of species diversity are ultimately generated by processes of speciation, extinction, and dispersal on evolutionary timescales [6]. The impact of these historical processes in different geographic regions or climates may lead to differences in global species diversity.

Traditionally, ecological studies focus on species distributions and environmental effects over short timescales [7,8,9], while evolutionary studies emphasize divergence times and diversification rates [10]. However, there is growing evidence that species richness patterns arise from a combination of environmental and evolutionary processes [11,12]. Phylogenetics provides a way to assess the importance of evolutionary history in determining patterns of species richness in terms of high or low rates of species diversification within geographic areas [13].

Various hypotheses have been proposed regarding species diversity distribution patterns. These hypotheses are not mutually exclusive and collectively influence species diversity in different ecosystems [14,15,16,17,18]. The energy hypothesis suggests that species richness is influenced by energy availability, with higher diversity in areas of high plant productivity, temperatures, precipitation, and evapotranspiration [19,20,21]. The habitat heterogeneity hypothesis posits that diverse habitats provide more ecological niches, promoting species differentiation and coexistence [22,23]. These hypotheses correspond to different factors influencing the patterns of species richness.

*Sorbus* L. (*sensu stricto*) is characterized by its pinnately compound leaves [24] and widely occurs in the temperate zone of the Northern Hemisphere. *Sorbus* species are also valued as ornamental plants due to their beautiful shapes, and brightly colored fruits and leaves. They are commonly used as street trees in European and other countries [25]. *Sorbus* also contributes to forest stability and soil and water conservation [26,27]. In recent years, research on *Sorbus* has focused on morphological characteristics and phylogeny [28,29], but the patterns and determinants of species richness in *Sorbus* are still unclear.

To fully understand the species richness pattern of *Sorbus*, we should use a combination of geographical distribution and phylogenetic data, considering the interaction between environmental factors and evolutionary history. In this study, we combined distribution data of *Sorbus* species with environmental predictors to analyze their global richness patterns and identify influencing factors. Using chloroplast genome information, we estimated the diversification rate of *Sorbus* within a phylogenetic framework and explored its diversity pattern through the interaction of environmental and evolutionary history.

## 2. Results

### 2.1. Global Patterns of Sorbus Species Richness

Species richness ranged from 1 to 31, with an average value of 2.14. Despite the widespread distribution of *Sorbus* species, most grid cells have low species richness (Table 1). Of the 110 *Sorbus* species, Asia has the highest species richness with 101 species, accounting for more than 90% of the total *Sorbus* species. The Hengduan Mountains (HDM) in East Asia are the primary distribution center, hosting 68 species. There are three grids with the highest species richness, with a maximum value of 31 (Figure 1). These grids are located within the HDM, including three provinces (Xizang, Yunnan, and Sichuan) in China, nine cities (autonomous prefectures), and 25 counties. The distribution of *Sorbus* species richness varies significantly among continents. In North America, there are six species, with five being endemic to the continent. Species richness in each grid cell does not exceed four. In Europe, there is a distribution of six species, four of which are endemic.

To determine spatial autocorrelation in *Sorbus* species richness, we calculated Moran’s I, which was 0.62 (*p* < 0.01), indicating a high degree of spatial autocorrelation in species richness (Figure 2). Globally, *Sorbus* species richness shows spatial clustering with high–high clusters concentrated in Asia (Figure 3), especially East Asia, indicating this region as the biodiversity differentiation center for *Sorbus*. High–low outliers present only in northern India.

The climatic predictors for the optimal growth range of the genus *Sorbus* were counted in each grid cell (Table 1). PWQ and RELE values were higher than averages, while the AET values were lower. Additionally, the optimum ranges for climate seasonality and RELE were narrow, indicating that 78 is suitable for high-altitude areas with relative rain–heat synchronization and stable climates.

### 2.2. Richness–Environment Relationships

At the global scale, the energy availability model had the highest adjusted R^2^ (adj. R^2^ = 19.83%), followed by the water availability model (adj. R^2^ = 18.93%), and the soil properties model had the lowest (adj. R^2^ = 3.54%) in a stepwise regression analysis (Table 2). The interpretation of different models varied across continents. In Asia, similar to the global results, the energy availability model (adj. R^2^ = 31.75%) best predicted species richness, while the soil properties model (adj. R^2^ = 7.27%) was the least predictive. In Europe, the energy availability model also had the highest explanation (adj. R^2^ = 26.01%), while the climate seasonality model had the lowest (adj. R^2^ = 10.16%). In North America, water availability (adj. R^2^ = 33.38%) was the most important variable, which was inconsistent with the others. The European soil properties model (adj. R^2^ = 24.97%) showed a high explanatory power, second only to energy availability. We used a set of factors to predict the relationship between richness and environmental factors. The model demonstrated low explanatory power and poor fit. However, combining all predictors into an integrated model improved the explanatory power significantly, with the highest adjusted R^2^ reaching 52.12% in the European region. Compared to the single group model, the adjusted R^2^ values in other regions also increased.

To evaluate the predictive power of the models, we calibrated four integrated models using global and continental data. We tested each set of data in different models and calculated the root mean square error (RMSE) (Figure 4). Testing in different models using data from Asia resulted in higher RMSE values than other continents. The RMSE calculated by the Asian data was much higher using European and North American data compared to other calibration models, indicating anomalies in the Asian data. We conducted stepwise regression analysis on five groups of predictors in the HDM. The adjusted R^2^ for each was higher than that of Asia, and the integrated model had an adjusted R^2^ of 90.91% (Table 3). In the HDM, the main influential variable was climate seasonality (adj. R^2^ = 49.26%), with high explanatory power. In Asia, soil properties had the least explanation, while, in the HDM, they were second only to climate seasonality (adj. R^2^ = 45.63%).

### 2.3. Analysis of Species Evolution Rates Using Complete Chloroplast Genomes

By using fossil information, we marked the differentiation time of each branch node in the phylogenetic tree of the genus *Sorbus* to obtain the maximum clade credibility (MCC) tree Appendix A. We calculated evolutionary rate and visualized the diversification rate graph for the genus *Sorbus* (Figure 5). During the late Eocene, *Sorbus* was in its initial differentiation stage, characterized by a generally high diversification rate. By the middle of the Miocene, the diversification rate had slowed down and reached a stable state. Speciation and extinction rates have steadily declined over the evolutionary history of the *Sorbus* genus, with only minor changes (Figure 6). Comparing the speciation rate and net diversification rate between the HDM and non-HDM regions, the HDM shows a slightly higher rate. Though the difference is very slight, the difference in net diversification rates increases over time from around 15 mya.

## 3. Discussion

### 3.1. The Relationship Between Climate and Species Richness Patterns

Stepwise regression analysis indicated that the adjusted R^2^ of the integrated models was higher than that of individual groups of predictors. This suggests that the species richness pattern of *Sorbus* species is influenced by a combination of environmental factors. Although there were large differences in *Sorbus* species richness across continents, the single group of environmental predictors that explained the most variation was energy or water availability. Similarly, our results support the energy–water hypothesis. Similar findings have been observed in other plants [30,31]. The energy–water hypothesis has important implications in defining species richness patterns along altitudinal gradients. Most species of the genus *Sorbus* were distributed in high-elevation areas, and climate and topography change with altitude, directly or indirectly affecting species richness. At higher altitudes, sufficient energy and precipitation are good for species diversity [32,33,34]. Many studies have supported the importance of energy, water, and their interactions on individual growth and distribution. However, the underlying physiological processes and mechanistic links between individual growth and general species richness patterns still require further exploration.

The species richness map indicates that the majority of *Sorbus* species are distributed in Asia. We separately analyzed the HDM, an area with high species richness in Asia. Combined with the clustering results, we believe that the HDM are the diversity center of *Sorbus.* In the HDM, climate seasonality is higher than other group predictors and other continents, indicating that climate seasonality is the most influential predictor of all predictors affecting species richness. At the same time, our results, together with previous research results, provide evidence for the climate seasonality enrichment hypothesis [35]. The climate in the HDM exhibits typical monsoon dynamics, with distinct rainy and dry seasons, hot and rainy summers, and cold and dry winters [36]. The contemporary geographical structure of the HDM has formed fold mountains and a series of fault basins due to plate tectonics, resulting in the formation of deep valleys [37,38]. The seasonal impact is more obvious in areas with high terrain relief [39]. When comparing soil characteristics in the HDM with those in other regions, the explanatory power of the HDM is significantly higher than that of other regions. The drastic environmental changes and similar effects of soil-forming factors in the HDM have led to significant soil diversity, covering multiple soil groups in soil classification [40,41]. Studies have shown that high levels of soil properties are often associated with high levels of biodiversity [42,43]. The HDM are one of the world’s biodiversity hotspots, creating a variety of habitats in a relatively small area [44]. We compared the adjusted R^2^ of habitat heterogeneity in Asia and the HDM, and the value of the HDM was slightly higher, indicating that habitat heterogeneity has a certain impact on species richness in the HDM.

Eastern Asia and eastern North America are similar in climate, vegetation type, and floristic composition, but plant species richness in East Asia is higher than in eastern North America [45]. The genus *Sorbus* is distributed in both East Asia and eastern North America, but the species diversity in the two regions differs greatly. Li reported that *Sorbus* originated in East Asia and spread from East Asia to North America twice between 26 mya and 16 mya [46]. During the migration process, the Bering Land Bridge may have provided a passage for it. East Asia has diverse environmental features and complex topography, and a monsoon climate. These conditions have promoted geographic isolation and allopatric speciation, providing refuge for many species, while the terrain of eastern North America is relatively flat [47]. Compared with eastern North America, East Asia was not as strongly affected by the Quaternary climate oscillation, which may have contributed to the lower level of extinction in East Asian lineages [48]. Therefore, the difference in species richness of *Sorbus* between East Asia and eastern North America suggests that environmental heterogeneity in East Asia may further promote speciation of *Sorbus* and be more conducive to the diversification of *Sorbus* species.

### 3.2. The Influence of Historical Climate on the Evolution Rate of Sorbus

The genus *Sorbus* may have originated in East Asia, with the crown group dating back to the Eocene period. The modern geographical distribution center is in the HDM. Around 41 mya, the Asian climate rapidly changed from a dry environment to a seasonally wet climate, resulting in the emergence of a subtropical monsoon climate in East Asia [49,50,51]. The unique topography and warm climate created complex ecological niches, leading to a high rate of speciation. Ultimately, it affected the global biodiversity hotspot of HDM and promoted biodiversity throughout East Asia. Therefore, this climatic and environmental influence may have promoted the increase in species diversity of the genus *Sorbus* in the late Eocene. In the early mid-Miocene, a cooling event occurred in the global ocean and atmosphere, and global temperatures dropped sharply. Climate cooling since the late Miocene may be the main reason for the decline in diversification rates.

### 3.3. Determinants of Global Species Diversity Patterns of Sorbus

Environmental factors play an important role in shaping global patterns of *Sorbus* species richness. Energy and water availability are the main environmental factors affecting species richness patterns of the genus *Sorbus*, and climate seasonality is the main environmental variable affecting the species diversity center (HDM). Our results show that the HDM have a high diversity of *Sorbus* species, but there is no significant difference between the diversification rates in the HDM and non-HDM, suggesting no correlation between diversification rates and species richness. Therefore, we suggest that mechanisms involved in speciation and extinction rates may not be drivers of global patterns of species richness in *Sorbus*. Our results are similar to those of Tietje [52], whose study demonstrated that species richness and diversification rates are not correlated geographically. Diversification rates can explain species richness on narrow spatial or phylogenetic scales, but there are still limitations in explaining global species richness patterns.

## 4. Materials and Methods

### 4.1. Species Distribution Data

Geographic distribution data for the genus *Sorbus* were obtained from herbarium specimens recorded in the Global Biodiversity Information Facility [53] (GBIF, https://www.gbif.org/, accessed on 5 July 2023). Species names were standardized according to Catalogue of Life (COL, http://www.catalogueoflife.org, accessed on 1 September 2023) and Plants of the World Online (POWO, https://powo.science.kew.org, accessed on 1 September 2023). For data accuracy, we verified each distribution record. For specimen records without coordinates, but with a detailed location description, we obtained latitude and longitude data on maps; we excluded records of introduced cultivation based on the genus *Sorbus*’s habits and records; there may be duplicate data for the same species. We removed duplicate records for the same species at the same location, and kept only one entry per species per location. The final dataset included 110 species and infraspecific units Appendix A.

The global administrative districts vector map was downloaded from the GADM database (https://gadm.org/data.html, accessed on 5 January 2024). These maps were converted into 200 × 200 km spatial grids using ArcGIS 10.8. Species distributions were also converted to grids with the same spatial resolution. To eliminate area effect on species richness estimates, we excluded incomplete grid cells located along state borders or in coastal areas with less than half land area. The final analysis included 3848 grid cells, with species richness defined as the total number of species present in each grid cell.

### 4.2. Environmental Data

To explore the impact of environmental factors on *Sorbus* species richness, we used 23 environmental predictors commonly employed in large-scale species diversity studies. These predictors were classified into five groups: environmental energy availability, water availability, climate seasonality, habitat heterogeneity, and soil properties (Table 4).

Energy availability predictors (MAT, MDR, ISO, MTWQ), water availability predictors (MAP, PWM, PWQ, PDQ), elevation, and climate seasonality data were downloaded from the WorldClim database (https://www.worldclim.org/, accessed on 5 January 2024) at a 2.5 arc-minute resolution. PET and AET were obtained from the Consortium of International Agricultural Research Centers (https://cgiarcsi.community/category/data/, accessed on 5 January 2024) at a 30 arc-second resolution. PETmin was calculated as monthly minimum PET of a year. RELE, RMAT, and RMAP were derived as ranges within each geographic unit. Soil properties were sourced from the Harmonized World Soil Database v1.2 (https://www.fao.org/soils-portal/soil-survey/soil-maps-and-databases/harmonized-world-soil-database-v12, accessed on 5 January 2024).

### 4.3. Statistical Analysis

First, we mapped the global distribution pattern of *Sorbus* species richness, using the Jenks function in ArcGIS 10.8 to divide species richness into five intervals. The largest graded intervals indicate areas with the highest species richness and corresponding suitable environmental conditions for the genus. Spatial autocorrelation was assessed using Moran’s I index, which ranges from −1 (perfect negative autocorrelation) to 1 (perfect positive autocorrelation), with 0 indicating randomness.

Stepwise regression analyses were conducted in R v4.3.2 to investigate the impact of environmental predictors on species richness. We used stepwise regression to explore the combined effects of environmental factors on species richness and ultimately identified a set of optimal predictors with the strongest explanatory power and minimal multicollinearity. We combined the selected groups of environmental predictors into an integrated model for analysis. The adjusted R^2^ coefficient was used to illustrate the model’s fit with higher values, with a higher value indicating a better fit. We built models based on species richness data and environmental predictors for each continent and applied these models to other continents to test their predictive power. We use the root mean square error to evaluate the prediction effect. A lower RMSE indicates higher predictive power and greater consistency in the richness–environment relationship between the predicted continent and the continent used to construct the model [54,55,56]. Statistical analysis was performed using the “glm” function and the “leaps” package in R v4.3.2.

### 4.4. Phylogenetic and Diversification Rate Analysis

We selected 39 materials of the genus *Sorbus*, with other genera of the Maleae and *Prunus mira* as outgroups. Three of these were newly sequenced and assembled chloroplast genome sequences of the genus *Sorbus*, and the other 36 chloroplast genome sequences and outgroups were retrieved from GenBank Appendix A.

The completed chloroplast genome sequences were aligned using MAFFT v.7 and adjusted appropriately. The aligned chloroplast genome sequences were imported into BEAUti v1.7.5 to set up taxa. The divergence time of the outgroup taxon *Prunus* from other clades was set to 73 mya using fossil calibration points [57]. The divergence time of *Sorbus* from the *Dichotomanthes* + *Osteomeles* clade was set at 51 mya based on the oldest *Sorbus*-like macrofossils [58]. The priors were set to normal, and the clock type was set to UCLN (Uncorrelated relaxed lognormal clock model). The Monte Carlo Markov Chain (MCMC) was run for 10,000,000 generations, with sampling every 1000 generations. The first 25% of the trees were discarded as burn-in to ensure that the chains were stationary. The convergence of the MCMC chain was monitored using Tracer v 1.7.2, ensuring that the effective sample size (ESS) values for each parameter were larger than 200.

The “setBAMMpriors” function in the “BAMMtools” package in R was used to generate prior parameters, and relevant parameters in the control file were adjusted accordingly. The MCMC was run for 10,000,000 generations, with sampling every 1,000 generations. The control file with set parameters was run in BAMM v2.5.0. The first 25% of samples were discarded as burn-in. The convergence of the results was checked in the “coda” package based on the MCMC sampling results to ensure that the ESS was larger than 200.

## 5. Conclusions

In conclusion, this study explores the global diversity distribution pattern of the *Sorbus* species and the impact of environment and evolutionary history on species richness patterns based on environmental predictors and diversification rates. According to the species diversity pattern, the HDM are the diversity center of the genus *Sorbus*. Our results support consistency between species richness and environment; the integrated model including all environmental predictors had the largest explanatory power for the species richness. Among the set of environmental predictors, energy and water availability were the main environmental factors affecting *Sorbus* species across continents. The difference is that the HDM are most affected by climate seasonality. Comparative results of diversification rates in the HDM and non-HDM indicate that diversification rates may not be the driving factor for the global species richness pattern of *Sorbus*.

## Figures and Tables

**Figure 1 plants-14-00338-f001:**
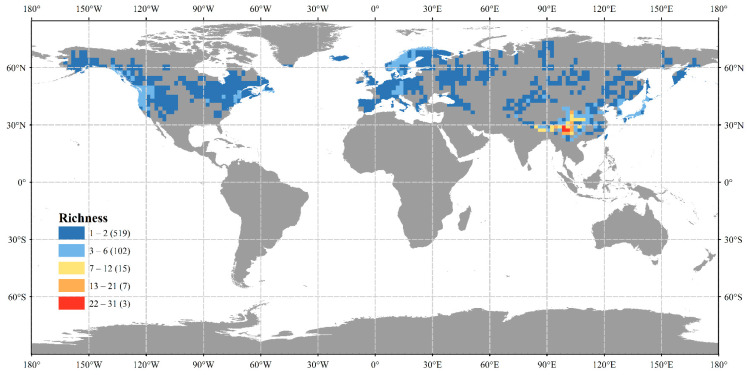
Global patterns of *Sorbus* species richness on each continent (numbers in brackets indicate how many grid cells there are under that richness class).

**Figure 2 plants-14-00338-f002:**
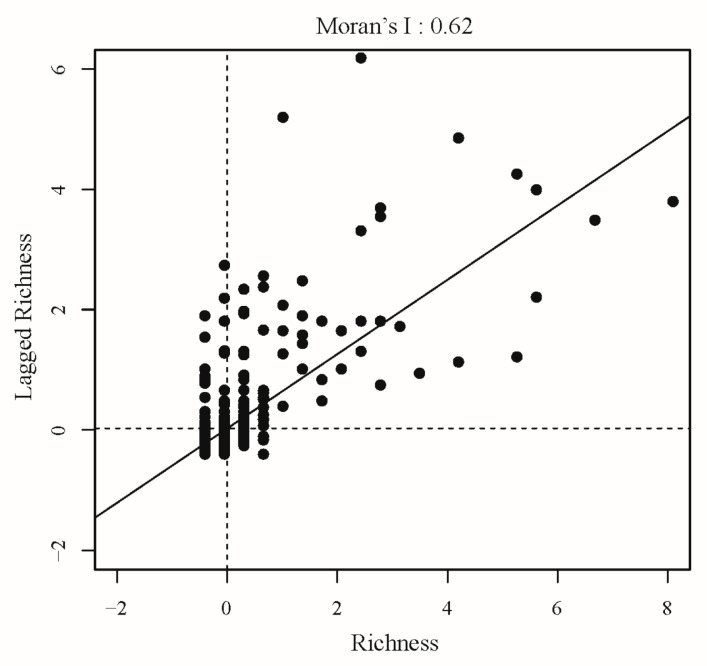
Global Moran’s I scatter plot of *Sorbus* richness.

**Figure 3 plants-14-00338-f003:**
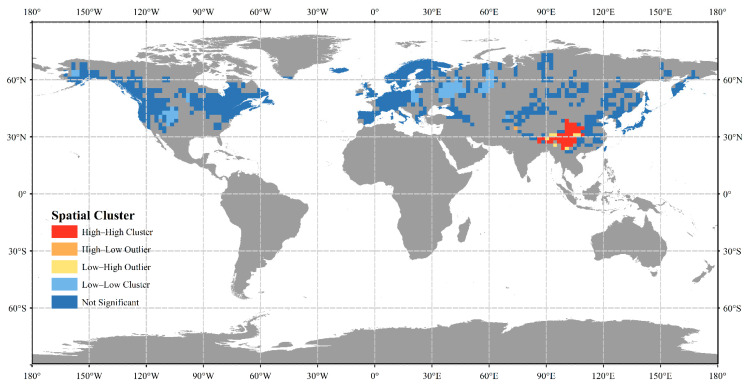
Spatial clustering of species richness patterns of *Sorbus* at global level.

**Figure 4 plants-14-00338-f004:**
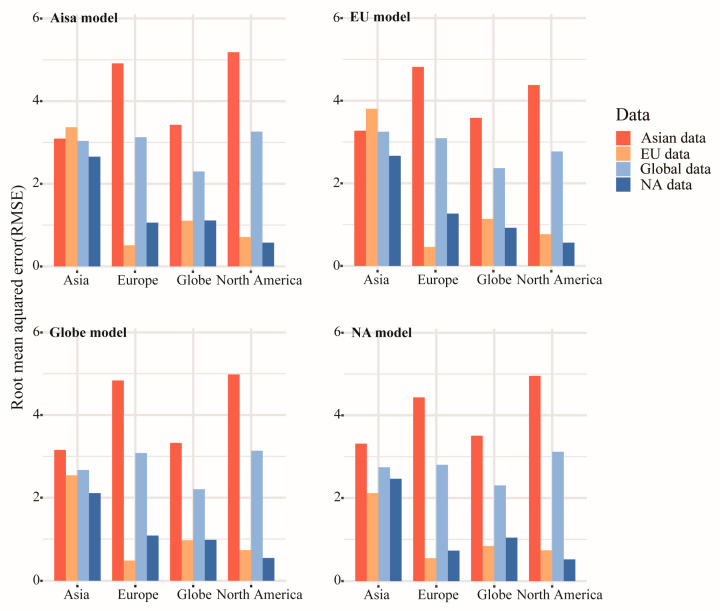
Root mean squared errors of the four integrated models were assessed when they were used to predict the *Sorbus* species richness at the global scale and in Asia, North America (NA), and Europe (EU) (color bars). Each of the four models was calibrated separately using Asian data, European data, North American data, and global data (x-axis).

**Figure 5 plants-14-00338-f005:**
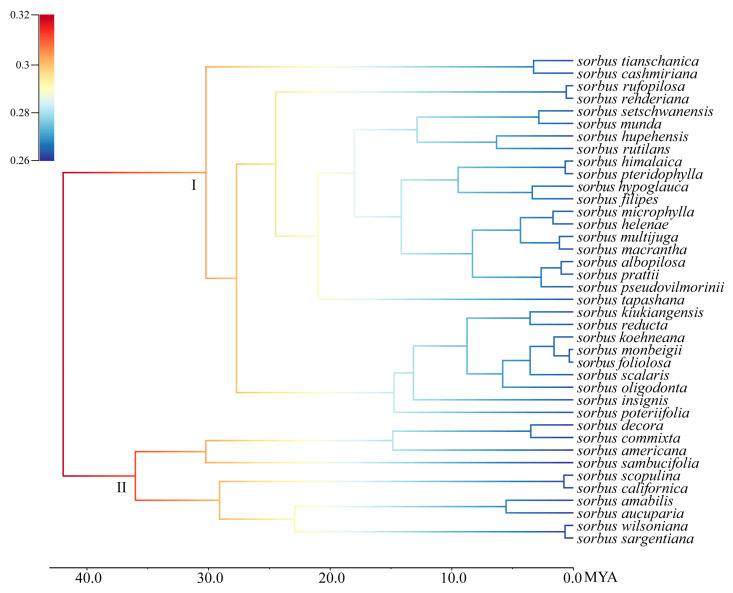
A phylorate plot showing the heterogeneity of diversification rates in the genus *Sorbus*. Clade I represents species distributed in the HDM, and clade II represents species distributed outside the HDM.

**Figure 6 plants-14-00338-f006:**
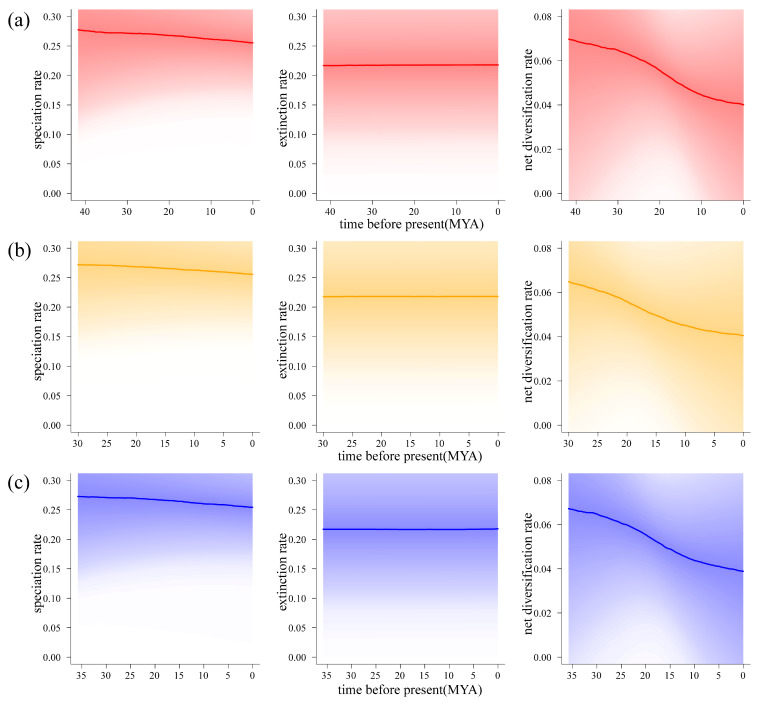
(**a**) Temporal variation in speciation, extinction, and net diversification rates for all species. (**b**) Temporal variation in speciation, extinction, and net diversification rates in the HDM. (**c**) Temporal variation in speciation, extinction, and net diversification rates outside the HDM.

**Table 1 plants-14-00338-t001:** Descriptive statistics (minimum, maximum, mean, standard deviation, optimal range) of species richness and environmental factors in *Sorbus*.

	Minimum	Maximum	Mean	Median	Standard Deviation	Optimal Range	
						Lower Limit	Upper Limit
Richness	1	31	2.14	1	2.82		
Mean annual temperature (MAT)	−13.17	20.30	4.37	4.11	6.43	2.13	11.09
Mean diurnal range (MDR)	4.80	17.30	10.56	10.40	2.50	9.56	11.44
Isothermality (ISO)	12.96	52.50	29.18	27.74	7.20	40.58	45.95
Mean temperature of the warmest quarter (MTWQ)	5.22	27.26	16.28	16.30	4.58	9.19	16.69
Potential evapotranspiration (PET)	327.65	2438.81	961.56	891.24	340.92	979.84	1265.50
Minimum monthly potential evapotranspiration (PETmin)	0.00	75.00	11.74	6.00	14.19	30.00	47.00
Mean annual precipitation (MAP)	35.70	2762.97	737.46	633.17	399.48	652.35	1280.60
Precipitation of wettest month (PWM)	7.11	583.89	113.69	96.17	69.06	137.46	241.64
Precipitation of driest quarter (PDQ)	2.13	376.77	92.19	76.96	68.60	24.41	59.94
Precipitation of warmest quarter (PWQ)	11.01	1491.57	259.11	225.89	162.30	350.25	634.82
Actual evapotranspiration (AET)	216.72	5909.95	984.34	720.13	838.78	601.39	811.05
Temperature seasonality (TSN)	346.23	2000.61	986.30	939.97	316.79	497.27	608.85
Annual range of temperature (ART)	15.89	65.89	37.44	36.73	9.55	23.57	27.68
Precipitation seasonality (PSN)	9.32	128.04	49.88	41.69	27.17	69.53	81.90
Range of elevation (RELE)	90.00	7355.00	1589.66	1352.00	1278.47	3301.00	3943.00
Range of mean annual temperature (RMAT)	1.36	41.63	9.02	7.05	6.83	17.75	21.74
Range of mean annual precipitation (RMAP)	39.00	4612.00	591.25	355.00	649.98	237.00	2774.00

**Table 2 plants-14-00338-t002:** The regression coefficients, *p*-values, and adjusted R^2^ (adj. R^2^) of stepwise regression models for each group of predictors and the integrated model for the globe and three continents.

	Globe			Asia			Europe			North America		
Groups	Predictors	Coefficients	adj. R^2^/%	Predictors	Coefficients	adj. R^2^/%	Predictors	Coefficients	adj. R^2^/%	Predictors	Coefficients	adj. R^2^/%
Energy Availability	ISO PET PETmin	0.1418 *** −0.0037 *** 0.0813 ***	19.83	MDR ISO PET	−0.4261 *** 0.3725 *** −0.0025 **	31.75	MAT MDR ISO MTWQ	0.4744 *** 0.5300 *** −0.1518 *** −0.5492 ***	26.01	MAT MDR ISO PETmin	0.0716 *** −0.1004 *** 0.0518 *** −0.0365 ***	25.30
Water Availability	MAP PWM PDQ PWQ	0.0088 *** −0.0353 *** −0.0336 *** 0.0064 ***	18.93	MAP PWM PDQ	0.0126 *** −0.0575 *** −0.0583 ***	18.75	PDQ	0.0052 ***	12.76	MAP PWM PDQ PWQ	0.0050 *** −0.0184 *** −0.0101 *** −0.0030 ***	33.38
Climate Seasonality	TSN PSN	−0.0033 *** 0.0327 ***	18.71	TSN	−0.0054 ***	23.65	TSN	−0.0011 ***	10.16	ART	−0.0406 ***	12.89
Habitat Heterogeneity	RMAP RELE	0.0004 * 0.0007 ***	14.22	RMAP RELE	0.0012 * 0.0022 **	17.70	RMAT RMAP RELE	0.1228 ** 0.0004 *** −0.0006 **	18.78	RMAP RELE	0.0006 *** −0.0003 *	23.97
Soil Properties	CLAY USDA TEB	0.1094 *** 0.2507 ** −0.0909 ***	3.54	CLAY pH_H_2_O TEB	0.1824 * 1.2888 * −0.2847 **	7.27	USDA OC pH_H_2_O TEB	0.2901 *** −0.0603 * −0.3845 ** 0.0471 *	24.97	OC TEB	0.0308 * −0.0458 ***	10.66
Integrated Model	PET MAP PWM PDQ PWQ RMAT	−0.0037 *** 0.0073 *** −0.0543 *** −0.0219 *** 0.0084 *** 0.1262 ***	37.60	MAT MDR ISO PET PWM pH_H_2_O	0.3824 *** −1.4080 *** 0.6102 *** −0.0088 *** −0.0531 *** 1.6000 ***	45.46	MAT MDR MTWQ AET ART PSN pH_H_2_O	−1.0537 *** 0.5921 *** 1.0413 *** −0.0005 *** −0.3483 *** −0.0376 *** −0.4294 ***	52.12	MTWQ PETmin PDQ TSN RMAP USDA OC	0.0905 *** −0.0252 *** −0.0078 ** −0.0049 ** 0.0005 *** 0.1115 ** 0.0393 **	48.20

Note: Significance levels: *** *p* < 0.001; ** *p* < 0.01; * *p* < 0.05.

**Table 3 plants-14-00338-t003:** The regression coefficients, *p*-values, and adjusted R^2^ (adj. R^2^) of stepwise regression models for each group of predictors and integrated model for Asia and the Hengduan Mountains.

	Asia			Hengduan Mountains		
Groups	Predictors	Coefficients	adj. R^2^/%	Predictors	Coefficients	adj. R^2^/%
Energy Availability	MDR ISO PET	−0.4261 *** 0.3725 *** −0.0025 **	31.75	MDR ISO	−2.5550 * 3.9930 *	37.85
Water Availability	MAP PWM PDQ	0.0126 *** −0.0575 *** −0.0583 ***	18.75	PDQ	−3.2750	32.24
Climate Seasonality	TSN	−0.0054 ***	23.65	PSN	−0.5172 **	49.26
Habitat Heterogeneity	RMAP RELE	0.0012 * 0.0022 **	17.70	RELE	0.4860 *	20.44
Soil Properties	CLAY pH_H_2_O TEB	0.1824 * 1.2888 * −0.2847 **	7.27	USDA OC pH_H_2_O TEB	−0.6272 ** −0.6338 ** 1.7820 *** −1.2580 *	45.63
Integrated Model	MAT MDR ISO PET PWM pH_H_2_O	0.3824 *** −1.4080 *** 0.6102 *** −0.0088 *** −0.0531 *** 1.6000 ***	45.46	PWM PWQ AET ART PSN RELE CLAY pH_H_2_O	7.4510 ** −7.5470 ** −1.1210 *** −1.7040 ** −0.4077 ** 1.0880 * 1.1220 * −1.2710 *	90.91

Note: Significance levels: *** *p* < 0.001; ** *p* < 0.01; * *p* < 0.05.

**Table 4 plants-14-00338-t004:** Environment predictors and their abbreviations used in the analyses.

Groups	Abbreviations	Environmental predictors
Energy Availability	MAT	Mean annual temperature (°C)
	MDR	Mean diurnal range (°C)
	ISO	Isothermality (°C)
	MTWQ	Mean temperature of the warmest quarter (°C)
	PET	Potential evapotranspiration (mm)
	PETmin	Minimum monthly potential evapotranspiration (mm)
Water Availability	MAP	Mean annual precipitation (mm)
	PWM	Precipitation of wettest month (mm)
	AET	Actual evapotranspiration (mm)
	PWQ	Precipitation of warmest quarter (mm)
	PDQ	Precipitation of driest quarter (mm)
Climate Seasonality	TSN	Temperature seasonality
	ART	Annual range of temperature (°C)
	PSN	Precipitation seasonality
Habitat Heterogeneity	RELE	Range of elevation (m) within each geographical unit
	RMAT	Range of mean annual temperature (°C) within each geographical unit
	RMAP	Range of mean annual precipitation (mm) within each geographical unit
Soil Properties	CLAY	Clay fraction (% wt.)
	USDA	The relative proportion of different grain sizes of mineral particles in soil
	OC	Organic carbon (% weight)
	pH_H_2_O	pH (−log(H^+^))
	TEB	Total exchangeable bases (cmol/kg)
	ECE	Electrical conductivity (dS/m)

## Data Availability

The distribution data of *Sorbus* were obtained from Global Biodiversity Information Facility (GBIF, https://www.gbif.org/ (accessed on 5 July 2023)). The environment predictors used in the study were obtained from WorldClim (https://www.worldclim.org/ (accessed on 5 January 2024)), Consortium of International Agricultural Research Centers (https://cgiarcsi.community/category/data/ (accessed on 5 January 2024)), and Harmonized World Soil Database v1.2 (https://www.fao.org/soils-portal/soil-survey/soil-maps-and-databases/harmonized-world-soil-database-v12/en/ (accessed on 5 January 2024)). Chloroplast genome sequences were retrieved from GenBank (https://www.ncbi.nlm.nih.gov/ (accessed on 27 March 2024)). The list and distribution of *Sorbus* species are in Appendix A.

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
