# Peer review of "Unraveling the Impact of Environmental Factors and Evolutionary History on Species Richness Patterns of the Genus Sorbus at Global Level"

_plants, 2025, doi:10.3390/plants14030338_

Round 1

Reviewer 1 Report

Comments and Suggestions for Authors

The manuscript “Phylogenomics analysis reveals the evolutionary history of Paleartic needle-leaved junipers” is appropriate to be published in Molecular phylogenetics and Evolution after considering a few aspects commented on in “Suggestion to authors”. The authors explore the global distribution patterns of species richness of the genus Sorbus. They use 23 environmental variables using stepwise regression with global and continental models, and an integrated model. In addition, they attempt to contrast two hypotheses that could explain the pattern of species richness distribution in this genus. I suggest accepting the manuscript but with a minor revision. I provide few comments to authors below.

Suggestion to authors:

I think the article is very interesting and very relevant, to study the distribution pattern of a very complex and difficult genus with a wide distribution, always is interesting. However, I think that the results and the discussion need to be explored a little more, especially because the most interesting part, which is the results for the HDM mountains, is not explored in depth. Looking at the maps of the distribution of the richness of sorbus species, it can be seen that their distribution is quite homogeneous and relatively low (the grids with the highest richness have 6 species in 200 km2) throughout the entire northern hemisphere. However, where there is a significant difference is in the HDM mountains grids, where they can reach 31 species. There is little depth, both in results and in discussion on this issue.

According to the results presented, the hypothesis of water and energy availability is the one that best explains the global distribution of sorbus richness, but what happens with the HDM mountains? There, the variables that have the greatest effect are the climate seasonality and the soil properties. Could the habitat heterogeneity hypothesis be the one that best explains this exceptional richness of species? In fact, much of the discussion of the results by the authors goes in this direction, when comparing the differences that explain the differences in species richness between East Asia and North America, when commenting on the characteristics of the HDM mountains where the orography of the territory stands out, etc.

Reading the article, I think it is true that the hypothesis of water and energy availability explains the general pattern, the genus prefers habitats with a high availability of these two factors, however the increase in species richness at a more local scale (HDM mountains) is marked by the other hypothesis, the heterogeneity hypothesis. I propose that the authors go deeper into this idea, accepting o refusing it and in the case of accept it reorder and adapt the text.

Other small suggestion are:

Page 2, line 58: Sorbus (sensu stricto): First, author should include the authors of the genus. Second, to understand what means “sensu stricto” they must indicate the taxonomic account followed.

Tables 1 and 4, I suggest to combine both tables and show in the results section. It could help to read and understand the manuscript. Currently, there are many abbreviations of the environmental factors along the manuscript, and their meaning are at the end of the manuscript.

Pages 2-3, lines 88-90: “Species richness ranged from 1 to 31, with an average value of 2.14. Despite the widespread distribution of Sorbus species, most grid cells have low species richness” This sentences talk about species richness and chould be included in the first paragraph of RESULTS.

Page 5, line 117: Only the result of the integrated model for Europe is shown. What about the other continents? And for the Global pattern?

Page 5, line 120. Include the meaning of the abreviation “RMSE”

Page 5, table 2: adj. R2: In the table is provided as % and in the text in decimal. Homogenize.

Page 8, line 164: Authors said that wáter is one of the main environmental variables affecting the species richness, however it is not commented in the results section. Include it.

Page 11, line 275: Authors should indicate how they do the selection of the variables for the integrated medels.

Page 11, lines 287-290. Authors should include the references of the calibration points (fossils

Finally should be interesting to include an small description of the soil properties of the HDM mountains in the introduction or in the discussion.

Author Response

Dear Reviewer:

Thank you for giving us the opportunity of reviewing our manuscript. Those comments are all valuable and very helpful for revising and improving our paper. Our replies to the reviewers comments are reported hereafter. All changes with respect to the previous submission have been highlighted in the Word file of the manuscript.

Comments 1: However, where there is a significant difference is in the HDM mountains grids, where they can reach 31 species. There is little depth, both in results and in discussion on this issue.

Response 1: Thank you for your valuable comments. We agree that the species richness in the Hengduan Mountains (HDM) grids (up to 31 species) is an important aspect of our findings. The list of species involved in the Hengduan Mountains is in Supplementary Material 1. Following your suggestion, we now include information on the provinces, cities, and counties involved in the analysis of the highest richness grids. In addition, in the Results and Discussion section, we analyze the factors that lead to higher species richness in the HDM, thus exploring this topic in more depth.

There are three grids with the highest species richness, with a maximum value of 31 (Figure 1). These grids are located within the HDM, including three provinces (Xizang, Yunnan, and Sichuan) in China, nine cities (autonomous prefectures) and 25 counties.

This change can be found on page 2, line 76-79.

Comments 2: According to the results presented, the hypothesis of water and energy availability is the one that best explains the global distribution of sorbus richness, but what happens with the HDM mountains?

Response 2: Thank you for your question. In our study, we found that the water and energy availability hypothesis best explained the global distribution of Sorbus species richness. However, when focusing on the Hengduan Mountains (HDM), the picture was more complicated. The explanatory power of water and energy availability was still high compared to Asia, but other factors, such as seasonal climate and soil properties, had relatively high explanatory power at the regional scale. Therefore, we elaborated on the environmental factors with high explanatory power in the Hengduan Mountains when analyzing and discussing them. Thank you very much for your careful reading of the manuscript.

Comments 3: Could the habitat heterogeneity hypothesis be the one that best explains this exceptional richness of species? However the increase in species richness at a more local scale (HDM mountains) is marked by the other hypothesis, the heterogeneity hypothesis. I propose that the authors go deeper into this idea, accepting o refusing it and in the case of accept it reorder and adapt the text.

Response 3: Thank you for your insightful suggestion. We appreciate your suggestion on whether the habitat heterogeneity hypothesis is the best explanation for the exceptional richness of species, which gave us new insights. In our results, habitat heterogeneity is not the highest in the Hengduan Mountains relative to other variables, but we do acknowledge that the explanatory power of habitat heterogeneity is relatively high compared to other regions (such as Asia). Therefore, we have discussed habitat heterogeneity in the manuscript, but have further elaborated on the habitat heterogeneity of the Hengduan Mountains.

The HDM is one of the world's biodiversity hotspots, creating a variety of habitats in a relatively small area [42]. We compared the adjusted R2 of habitat heterogeneity in Asia and the HDM, and the value of the HDM was slightly higher, indicating that habitat heterogeneity has a certain impact on species richness in the HDM.

This change can be found on page 9, line 193-198.

Comments 4: Page 2, line 58: Sorbus (sensu stricto): First, author should include the authors of the genus. Second, to understand what means “sensu stricto” they must indicate the taxonomic account followed.

Response 4: Thank you for your valuable suggestions. Sorbus sensu lato has been divided into five or six genera(Aria (Pers.) Host, Chamaemespilus Medik., Cormus Spach, Micromeles Decne., Sorbus L.(sensu stricto)  and Torminalis Medik.), and the genus Sorbus s.s. is restricted to species with pinnately compound leaves. We have now included the authors of the genus, explained the definition of “Sensu stricto”, and added relevant references. Sorbus L. (sensu stricto) is characterized by the pinnately compound-leaves [24], and widely occurs in the temperate zone of the Northern Hemisphere.

This change can be found on page 2, line 56.

Comments 5: Tables 1 and 4, I suggest to combine both tables and show in the results section. It could help to read and understand the manuscript. Currently, there are many abbreviations of the environmental factors along the manuscript, and their meaning are at the end of the manuscript.

Response 5: Thank you for your valuable suggestions on the tables. We understand that merging the tables may improve readability, but since Table 4 contains long environmental variable names, merging the tables may affect the clarity and readability of the table. Therefore, we decided to keep Table 1 and Table 4 separate. However, we added a column to Table 1 to clearly list the group to which each variable belongs.

This change can be found on page 5, Table1.

Comments 6: Pages 2-3, lines 88-90: “Species richness ranged from 1 to 31, with an average value of 2.14. Despite the widespread distribution of Sorbus species, most grid cells have low species richness” This sentences talk about species richness and chould be included in the first paragraph of RESULTS.

Response 6: Thank you for pointing this out. We have moved this sentence to the first paragraph of the results.

This change can be found on page 2, line 72-73.

Comments 7: Page 5, line 117: Only the result of the integrated model for Europe is shown. What about the other continents? And for the Global pattern?

Response 7: Thank you for your suggestion. The European integrated model has the highest explanatory power, so we describe it in the Results section. We carefully considered your suggestion and have compared the results for other continents and global models with the single set variables in the manuscript: However, combining all variables into an integrated model improved the explanatory power significantly, with the highest adjusted R2 reaching 52.12% in the European region. Compared to the single group model, the adjusted R2 values in other regions also increased.

This change can be found on page 5, line 116-119.

Comments 8: Page 5, line 120. Include the meaning of the abreviation “RMSE”

Response 8: Thank you for the reviewer's valuable comments. We have added the meaning of the abbreviation 'RMSE' in the revised manuscript: We tested each set of data in different models and calculated the root mean square error (RMSE) (Figure 4).

This change can be found on page 5, line 122.

Comments 9: Page 5, table 2: adj. R2: In the table is provided as % and in the text in decimal. Homogenize.

Response 9: Thank you for pointing out this detail. We have used % in the manuscript and have unified the representation in tables and text to ensure consistency.

This change can be found on page 5-6, line 104-133.

Comments 10: Page 8, line 164: Authors said that wáter is one of the main environmental variables affecting the species richness, however it is not commented in the results section. Include it.

Response 10: Thank you for pointing this out. In the manuscript Results we analyzed water availability in North America: In North America, water availability (adj. R2 = 33.38%) was the most important variable, which was inconsistent with the others. Following your suggestion we made the following additions to the Results: At the global scale, the energy availability model had the highest adjusted R2 (adj. R2 = 19.83%), followed by the water availability model (adj. R2 = 18.93%).

This change can be found on page 4, line 105 and 111-112.

Comments 11: Page 11, line 275: Authors should indicate how they do the selection of the variables for the integrated models.

Response 11: Thank you for your valuable suggestions. We have explained how the integrated model was selected. We combined the selected groups of environmental variables into an integrated model for analysis.

This change can be found on page 11, line 291-292.

Comments 12: Page 11, lines 287-290. Authors should include the references of the calibration points (fossils

Response 12: Thank you for your valuable suggestions. We have added the corresponding references of the calibration points (fossils) as suggested.

  1. Lo, E.Y.Y.; Donoghue, M.J. Expanded phylogenetic and dating analyses of the apples and their relatives (Pyreae, Rosaceae). Mol. Phylogenet. Evol. 2012, 63, 230–243, https://doi.org/10.1016/j.ympev.2011.10.005.
  2. Vikulin, S.V.; Bystriakova, N.; Schneider, H.; Jolley, D. Plant macrofossils from Boltysh crater provide a window into early Cenozoic vegetation. In Volcanism, Impacts, and Mass Extinctions: Causes and Effects; Keller, G., Kerr, A.C.; Geological Society of America: Boulder, USA, 2014; Volume 505, pp. 147–169.

This change can be found on references (54, 55), page 15, line 466-470.

Comments 13: Finally should be interesting to include an small description of the soil properties of the HDM mountains in the introduction or in the discussion.

Response 13: Thank you for your insightful suggestion! We briefly describe the soil properties of the HDM in the Discussion section and relate this to the high degree of soil properties explaining environmental predictors.

When comparing soil characteristics in the HDM with those in other regions, the explanatory power of the HDM is significantly higher than that of other regions. The drastic environmental changes and similar effects of soil-forming factors in the HDM have led to significant soil diversity, covering multiple soil groups in soil classification [40,41]. Studies have shown that high levels of soil properties are often associated with high levels of biodiversity [42,43].

This change can be found on page 9, line 193-198.

Reviewer 2 Report

Comments and Suggestions for Authors

Dear authors, 

I gave a check on your manuscript, which I was expecting to be quite interesting, given the title (which should be slightly changed, in my opinion, since it is poorly written, and should be something like: Unraveling the Impact of Environmental Factors and Evolutionary History on Species Richness and Distribution Patterns of the genus Sorbus at global level). I checked the abstract, which is quite confusing (I am providing you a thorough review for it here below). 

Then, as I normally do for any manuscript, after a general overview (which made me think that the person which wrote the abstract is somebody else than the authors, since the rest of the manuscript is certainly written quite better), I focused on the Data and Methods section, which begins with the following sentence (lines 229-231): “Geographic distribution data for the genus Sorbus were obtained from herbarium specimens recorded in the Global Biodiversity Information Facility (GBIF, https://www.gbif.org/, accessed on 5 July 2023)”. The data in the GBIF are available to researchers worldwide under a CC 4.0 BY-NC license. This means that the datasets which are obtained from the GBIF must (MUST) be cited properly. When you download the data, a DOI is provided. This MUST be cited in the manuscript. Otherwise, you are stealing data from those who worked (sometimes for a whole life) for collecting and publishing them. If something is available for free, it does not mean that it is available under no restrictions. Furthermore, GBIF DOIs allow everybody to download the same dataset in order to replicate your experiments, and this is fundamental for modern science.

Unless the authors address this issue, the manuscript should be rejected. 

I do not deem necessary to review further the manuscript until a resubmission is done.

Here there are some comments on the abstract

Lines 11-13: The authors wrote: “Understanding the distribution patterns of species richness is a fundamental question in evolutionary biology and ecology. Here, we comprehensively assessed the role of these factors in shaping the global distribution pattern of Sorbus species richness.

This two sentences are quite problematic, and internally conflicting. The first one talks about “distribution patterns of species richness”, while the second talks about global distribution patterns. This is quite confusing., since the title talks about species richness and distribution patterns as two distinct things to understand and investigate. Furthermore, the second sentence here talks about some “factors”, which are not introduced before. Even if it is an abstract, it should be more informative and less confusing.

Line 14-15: “We used the specimen recorded to divide the global continent into 200×200 km grids, and map the richness of 110 Sorbus species”. What is the “global continent”?

Lines 15-17: “We analyzed five groups of 23 environmental variables using stepwise regression with global and continental models, and an integrated model.” This is the abstract, and should allow the readers to understand generally the methodology which was applied in the study. This sentence is quite complicate, since if th authors state they “grouped” the 23 environmental predictors (not variables), they should also state how they did it. Or better, avoid saying that they grouped them, in the abstract, and describe the process properly in the data and methods..

Lines 18-20: “Based on chloroplast genome data and the fossil record, we constructed a phylogenetic tree and estimated Sorbus diversification rate. Results show that energy-water availability is the main driver of global Sorbus diversity”. OK, now I am at a loss. The authors till line 18 were describing a modeling approach, and now a line about a phylogenetic tree pops out, just to be immediately forgotten, going back to the results of the model…

Lines 19-30: I can extract some information from these lines, but they are written poorly. In general, the whole abstract is confusing, and should be rewritten thoroughly.

Best regards

Author Response

Thank you for giving us the opportunity of reviewing our manuscript. Those comments are all valuable and very helpful for revising and improving our paper. Our replies to the reviewers comments are reported hereafter. All changes with respect to the previous submission have been highlighted in the Word file of the manuscript.

Comments 1: This means that the datasets which are obtained from the GBIF must (MUST) be cited properly.

Response 1: Thank you for your careful review and pointing out the critical issues regarding the correct citation of GBIF data. We sincerely appreciate your attention to detail and your rigor in data usage and citation standards in scientific research. We fully understand the importance of correctly citing datasets, especially those obtained from repositories such as GBIF, which are made available through a lot of effort by researchers. As you correctly pointed out, the DOI of the datasets must be properly cited to ensure the transparency and reproducibility of our research.

We have added the inclusion of the relevant DOI of the GBIF datasets in the revised manuscript, which is essential to ensure scientific rigor and openness. Thank you again for your careful consideration and pointing out this important aspect of the manuscript.

53.GBIF.org. GBIF occurrence download. 2023. https://doi.org/10.15468/dl.2n4ze8.

This change can be found on page 10 and 15, line 246 and 465.

Comments 2: which should be slightly changed, in my opinion, since it is poorly written, and should be something like: Unraveling the Impact of Environmental Factors and Evolutionary History on Species Richness and Distribution Patterns of the genus Sorbus at global level

Response 2: Thank you for your valuable suggestion on our title. We have revised it to make it clearer and more accurate. The revised title is: "Unraveling the Impact of Environmental Factors and Evolutionary History on Species Richness Patterns of the Genus Sorbus at Global Level "We believe that this revised title can better convey the core content of the article.

This change can be found on page 1, line 2-4.

Comments 3: Lines 11-13: The authors wrote: “Understanding the distribution patterns of species richness is a fundamental question in evolutionary biology and ecology. Here, we comprehensively assessed the role of these factors in shaping the global distribution pattern of Sorbus species richness.” This two sentences are quite problematic, and internally conflicting. The first one talks about “distribution patterns of species richness”, while the second talks about global distribution patterns. This is quite confusing., since the title talks about species richness and distribution patterns as two distinct things to understand and investigate. Furthermore, the second sentence here talks about some “factors”, which are not introduced before. Even if it is an abstract, it should be more informative and less confusing.

Response 3: Thank you for your valuable comments. We appreciate your insightful comments on the issues in the Abstract. We did not clarify the logic of the first two sentences of the Abstract, which caused some confusion. We realize that these sentences can be clearer and less confusing. We have revised these questionable sentences and changed them to species richness patterns to make them consistent. The "factors" that were not introduced before have also been deleted.

Here is the revised section:

Understanding the drivers of species richness patterns is a major goal of ecology and evolutionary biology, and the drivers vary across regions and taxa. Here, we assessed the influence of environmental factors and evolutionary history on the pattern of species richness in the genus Sorbus (110 species).

This change can be found on page 1, line 11-14.

Comments 4: Line 14-15: “We used the specimen recorded to divide the global continent into 200×200 km grids, and map the richness of 110 Sorbus species”. What is the “global continent”?

Response 4: Thank you for pointing out the ambiguity of the term "global continents". We agree that "global continents" is not a standard or clear expression. In our study, we intended to refer to the entire landmass or continents of the Earth. Therefore, we have revised that part of the question: We mapped the global species richness pattern of Sorbus at a spatial resolution of 200x200 km, using 10,652 specimen records. We appreciate your attention to this detail and hope that this revision will clarify the meaning.

This change can be found on page 1, line 14-15.

Comments 5: Lines 15-17: “We analyzed five groups of 23 environmental variables using stepwise regression with global and continental models, and an integrated model.” This is the abstract, and should allow the readers to understand generally the methodology which was applied in the study. This sentence is quite complicate, since if th authors state they “grouped” the 23 environmental predictors (not variables), they should also state how they did it. Or better, avoid saying that they grouped them, in the abstract, and describe the process properly in the data and methods.

Response 5: Thank you for your valuable suggestions. We agree with you that the process of grouping environmental predictors should be explained in more detail in the Materials and Methods section rather than in the Abstract. We have made modifications to avoid the complexity of reading and describe the methods in a concise manner.

In the Materials and Methods section, we will describe the classification of environmental predictors in detail.

In your suggestion to use "environmental predictors" instead of "environmental variables", this change is more in line with our study and we have made the changes in the manuscript. We are very grateful for your professional suggestions.

Comments 6: Lines 18-20: “Based on chloroplast genome data and the fossil record, we constructed a phylogenetic tree and estimated Sorbus diversification rate. Results show that energy-water availability is the main driver of global Sorbus diversity”. OK, now I am at a loss. The authors till line 18 were describing a modeling approach, and now a line about a phylogenetic tree pops out, just to be immediately forgotten, going back to the results of the model…

Response 6: Thank you for your valuable comments. We appreciate your observation about the abrupt transitions between parts of the statement. We agree that such transitions can be confusing because they are not clearly connected. To address this, we have revised the section in question to provide a clearer transition between the two parts. We thank you for your careful and thoughtful reading of the manuscript.

Here is the revised section:

We used stepwise regression to assess the relationship between 23 environmental predictors and species richness and estimated the diversification rate of Sorbus based on chloroplast genome data. The effects of environmental factors were explained by adjusted R², and evolutionary factors were inferred based on differences in diversification rates.

This change can be found on page 1, line 15-19.

Comments 7: Lines 19-30: I can extract some information from these lines, but they are written poorly. In general, the whole abstract is confusing, and should be rewritten thoroughly.

Response 6: Thank you for your valuable suggestions. We greatly appreciate your comments on the Abstract. Based on your suggestions, we have completely rewritten the Abstract to improve its structure and coherence.

The following is the revised Abstract:

Understanding the drivers of species richness patterns is a major goal of ecology and evolutionary biology, and the drivers vary across regions and taxa. Here, we assessed the influence of environmental factors and evolutionary history on the pattern of species richness in the genus Sorbus (110 species). We mapped the global species richness pattern of Sorbus at a spatial resolution of 200x200 km, using 10,652 specimen records. We used stepwise regression to assess the relationship between 23 environmental predictors and species richness and estimated the diversification rate of Sorbus based on chloroplast genome data. The effects of environmental factors were explained by adjusted R², and evolutionary factors were inferred based on differences in diversification rates. We found that the species richness of Sorbus was highest in the Hengduan Mountains (HDM), forming a biodiversity center. Among the selected environmental predictors, the integrated model including all environmental predictors had the largest explanatory power for the species richness. The determinants of species richness show regional differences. At the global and continental scale, energy and water availability becomes the main driving factor. In contrast, climate seasonality is the primary factor in the HDM. The diversification rate results showed no significant differences between HDM and non-HDM, suggesting that evolutionary history may have limited impact on the pattern of Sorbus species richness. We conclude that environmental factors play an important role in shaping the global pattern of Sorbus species richness, while diversification rates have a lesser impact.

We believe that this revision provides a clearer and more organized overview of our findings. We hope that the reorganization and rephrasing address the issues you raised and that the Abstract is now more intuitive for readers. We sincerely thank you for your careful review and helpful suggestions, which helped improve the manuscript.

This change can be found on page 1, line 11-29.

Round 2

Reviewer 1 Report

Comments and Suggestions for Authors

Congratulations, I have no further suggestions to add to the authors in this second revision.

Author Response

Dear Reviewer,

Thank you for your positive feedback. We really appreciate your time and effort in reviewing our manuscript. We are glad that our revisions met your expectations and appreciate your constructive comments throughout the process.

Reviewer 2 Report

Comments and Suggestions for Authors

Dear authors,

I have read with interest the revised version of your manuscript.

First of all, thank you for properly citing the data you downloaded from the GBIF.

I have further comments, especially regarding the Results section:

Please, rearrange the position of table 1. It is cited in the text before figure 1, but it is placed after figure 3.

Line 87: “the biodiversity differentiation center for Sorbus plant”. Plant? It is a genus…

Line 90: “in each grid cell of the delineation”. Delineation? What? 

The caption of table 1 is poor, and it seems to be missing a part. Tables and figures together with their captions should be self-explicative. Thus, you should provide in the caption a description of the content of the table/figure, and an explanation of all of the abbreviations you use in them. Plus, table 1 has something strange… Should the first row (richness) be separated from the three groups?Please, put a separator there, otherwise it seems that you calculated then richness for environmental energy availability only.

Lines 95-96, caption to figure 1: “species composition”? Actually the map shows species richness, not composition. To depict the composition, you should list all of the infragenric taxa which are present in each continent.

Lines 98-99: the caption is too poor. Tables and figures together with their captions should be self-explicative. Thus, you should provide in the caption a description of the content of the table/figure, and an explanation of all of the abbreviations you use in them.

Line 101: the caption is too poor. Tables and figures together with their captions should be self-explicative. Thus, you should provide in the caption a description of the content of the table/figure, and an explanation of all of the abbreviations you use in them.

Lines 120-1: “comprehensive models”. What is a comprehensive model? A model is a model…

Lines 122-4: this sentence is difficult to understand, without reading the following part of the paragraph. In general, this part should be rewritten stating before how the models were done, what is the aim of the comparison, and how it was done, explaining that the predictive power od each model done for a continent was tested on the continents (if I understood correctly what the authors wrote). Plus, I would like to have at least some references on the use of RMSE asa metric for the predictive power of the models

Figure 4 is too small, and hardly readable.

The caption of table 3 is poor, and it seems to be missing a part. Tables and figures together with their captions should be self-explicative. Thus, you should provide in the caption a description of the content of the table/figure, and an explanation of all of the abbreviations you use in them. 

Figure 5 should be rearrange and widened, to improve readability. The scale could be rotated of 90° and placed at the bottom or at the top, so that the image could be widened.

Figure 6 could be widened at least to full page to improve readability.

As far as the rest of the manuscript, I have no particular issues, but the fact that the English could be improved, and requires a thorough revision by a native English speaker.

Some examples:

  • Lines 20-1 “the species richness of Sorbus was highest in the Hengduan Mountains (HDM), forming a biodiversity center”. This should be rephrased. Instead of “forming a biodiversity center” the authors should write something as “which is probably the biodiversity differentiation center of the genus”, or something like this. A “biodiversity center” is a term which is normally used to define an institution that studies biodiversity.
  • Line 24: “becomes” should be “become

In general, an English review could improve the readability of the manuscript, which in any case is fully understandable.

Best regards

SM

Author Response

Manuscript. Number.: plants-3404101

Title: Unraveling the Impact of Environmental Factors and Evolutionary History on Species Richness Patterns of the Genus Sorbus at Global Level

Thank you for giving us the opportunity of reviewing our manuscript. Those comments are all valuable and very helpful for revising and improving our paper. Our replies to the reviewers comments are reported hereafter. All changes with respect to the previous submission have been highlighted in the Word file of the manuscript.

Comments 1: Please, rearrange the position of table 1. It is cited in the text before figure 1, but it is placed after figure 3.

Response 1: Thank you for your corrections, which ensured that the structure of the article was clearer and more coherent. We have adjusted the position of Table 1 according to your suggestion, so that it appears before Figure 1, which is consistent with the order of citation in the text.

This change can be found on page 3, line 94-95.

Comments 2: Line 87: “the biodiversity differentiation center for Sorbus plant”. Plant? It is a genus…

Response 2: Thank you for your correction. We have changed the original text from “the biodiversity differentiation center for Sorbus plant” to “the biodiversity differentiation center for the Sorbus ” to express the meaning more accurately.

This change can be found on page 2, line 87.

Comments 3: Line 90: “in each grid cell of the delineation”. Delineation? What?

Response 3: Thank you for your valuable feedback. The word "delineation" you mentioned can indeed cause confusion. Here, "delineation" refers to the division of the 200x200 grid that we made. To avoid confusion, we have changed the statement to "in each grid cell". Hopefully, this change will make this section clearer and easier to understand.

This change can be found on page 3, line 90.

Comments 4: The caption of table 1 is poor, and it seems to be missing a part. Tables and figures together with their captions should be self-explicative. Thus, you should provide in the caption a description of the content of the table/figure, and an explanation of all of the abbreviations you use in them. Plus, table 1 has something strange… Should the first row (richness) be separated from the three groups?Please, put a separator there, otherwise it seems that you calculated then richness for environmental energy availability only.

Response 4: Thank you for your insightful suggestions. . We have revised the title to be more descriptive and self-explanatory. The new title is: "". We have added explanations for all abbreviations used in the table to ensure that the table is fully understandable to the reader. We have added separators between the richness data and the three environmental groups.

This change can be found on page 3, line 94-95.

Comments 5: Lines 95-96, caption to figure 1: “species composition”? Actually the map shows species richness, not composition. To depict the composition, you should list all of the infragenric taxa which are present in each continent.

Response 5: Thank you for your valuable feedback. You pointed out that the term “species composition” in the title of Figure 1 is a misnomer. Indeed, Figure 1 shows species richness rather than species composition. We have changed the title to “Figure 1. Global patterns of Sorbus species richness on each continent (Numbers in brackets indicate how many grid cells there are under that richness class). ” to more accurately describe the content of the figure.

Comments 6: Lines 98-99: the caption is too poor. Tables and figures together with their captions should be self-explicative. Thus, you should provide in the caption a description of the content of the table/figure, and an explanation of all of the abbreviations you use in them.

Response 6: Thank you for your valuable suggestions. Based on your comments, we have modified the title of Figure 2 to ensure that it is clearer and more self-explanatory. We changed the title to “Global Moran’s I scatter plot of Sorbus species richness”.

This change can be found on page 4, line 100.

Comments 7: Line 101: the caption is too poor. Tables and figures together with their captions should be self-explicative. Thus, you should provide in the caption a description of the content of the table/figure, and an explanation of all of the abbreviations you use in them.

Response 7: Thank you for your valuable comments. Based on your suggestions, we have modified the title of Figure 3 to make it clearer and more self-explanatory. The new title is " Spatial clustering of species richness patterns of Sorbus at global level.", which is intended to more accurately reflect what is presented in the figure.

This change can be found on page 5, line102.

Comments 8: Lines 120-1: “comprehensive models”. What is a comprehensive model? A model is a model…

Response 8: Thank you for your comments. What we refer to as an ‘ensemble model’ is the integration of selected environmental predictors (such as energy, water, habitat heterogeneity, etc.) into a comprehensive analytical model. By integrating multiple predictors, we are able to improve the accuracy and predictive power of the model while taking into account the interrelationships between the individual variables.

Comments 9: Lines 122-4: this sentence is difficult to understand, without reading the following part of the paragraph. In general, this part should be rewritten stating before how the models were done, what is the aim of the comparison, and how it was done, explaining that the predictive power od each model done for a continent was tested on the continents (if I understood correctly what the authors wrote). Plus, I would like to have at least some references on the use of RMSE asa metric for the predictive power of the models

Response 9: Thank you for your valuable comments. In response to your suggestions on the understanding and structure of lines 122-124, we have reorganized the statements and added them to the Materials and methods section. Specifically, we clarified the process of building each model before explaining the purpose of model building and comparison, and explained how to use the root mean square error (RMSE) to evaluate the predictive ability of the model. In addition, we have added relevant literature to support RMSE as a common indicator for evaluating prediction performance.

The revised section is as follows:

We built models based on species richness data and environmental predictors for each continent and applied these models to other continents to test their predictive power. We use the root mean square error to evaluate the prediction effect. A lower RMSE indicates higher predictive power and greater consistency in the richness- environment relationship between the predicted continent and the continent used to construct the model [54–56].

This change can be found on page 13, line 290- 295.

Comments 10: Figure 4 is too small, and hardly readable.

Response 10: Thank you for your valuable feedback. We have adjusted Figure 4 to ensure that the content is clearer and more readable.

This change can be found on page 7, line 134.

Comments 11: The caption of table 3 is poor, and it seems to be missing a part. Tables and figures together with their captions should be self-explicative. Thus, you should provide in the caption a description of the content of the table/figure, and an explanation of all of the abbreviations you use in them.

Response 11: Thank you for your valuable comments. Based on your suggestions, we have revised and improved the title of Table 3 to more clearly describe the table content and explain all the abbreviations used. The revised title is: Table 3. The regression coefficients, p-values and adjusted R2 (adj. R2) of stepwise regression models for each group of predictors and integrated model for Asia and the Hengduan Mountains.

This change can be found on page 7, line 139-140.

Comments 12: Figure 5 should be rearrange and widened, to improve readability. The scale could be rotated of 90° and placed at the bottom or at the top, so that the image could be widened.

Response 12: Thank you for your helpful suggestions on Figure 5. We have moved the scale bar to the left of the image and widened the image to improve its clarity and overall presentation.

This change can be found on page 8, line 154.

Comments 13: Figure 6 could be widened at least to full page to improve readability.

Response 13: Thank you for your comments on Figure 6. We have expanded the figure, which I believe improves its clarity and makes it easier for readers to follow the details.

This change can be found on page 9, line 158.
